# The Role of Stakeholders in Creating Mobility in Logistics Systems of Polish Cities

Edyta Przybylska, Marzena Kramarz and Katarzyna Dohn *

Faculty of Organization and Management, Silesian University of Technology, Roosevelt 26-28 Str., 41-800 Zabrze, Poland
* Correspondence: katarzyna.dohn@polsl.pl; Tel.: +48-322-777-407

**Abstract:** The basis for developing logistics solutions in cities is to know the requirements and expectations of current and potential transport users and for decision makers to strive to meet them. In building an urban logistics strategy, it is therefore necessary to take into account different stakeholder groups. Building stakeholder relations should be aimed at involving them in the development of a sustainable transport policy for the city. It should be noted, however, that the importance of stakeholders in transport policy is diverse. This assumption was made in the paper, which aimed to identify the role of the various groups of urban logistics stakeholders in the sustainable movement of people. This challenge is subordinated to the methodology proposed in the paper, which combines the analysis of urban logistics stakeholders and the assignment of roles to them in the pursuit of sustainable flows of people in the city with the identification of tasks in which stakeholders characterized by a particular role should be involved. Achieving the stated goal required collaboration with experts. Research on the roles of stakeholders, as well as the tasks in which they should be involved, was conducted in Polish cities. Infrastructure managers, small mobility organisers, public safety organisations and public transport organisations turned out to be the development leaders who, as a result, were recommended to be included in most tasks by local government units of Polish cities. What deserves special attention in the results obtained, on the other hand, is the role of the unpredictable main player, which is other cities.

**Keywords:** urban logistics; stakeholders; sustainable urban transport

## 1. Introduction

The continuous development of modern cities is generating an increasing need for people to move, among other reasons, in order to move away from their place of residence to their place of work, to study and to meet social and cultural needs. The fulfilment of these needs is seen, inter alia, through the modes of movement (walking, public transport, micromobility means and individual cars). Each of these forms is complementary and influences the others. Nevertheless, the first three form the basis of a system that aims to encourage city dwellers and visitors to give up travelling by private car. In order to achieve the intended effects in this respect, urban mobility should be supported by changes in road transport, the use of intelligent public transport and traffic systems, and the promotion of multimodality, i.e., supporting the combination of different modes of transport and their integration. The concept of sustainable urban mobility consists of [1]:

- Activities that lead to the creation of transport systems that meet the needs of all users of urban transport systems;
- Balancing and meeting transport needs;
- Integration of modes of transport;
- Ensuring sustainable urban development;
- Cost optimization;
- Increasing the use of existing transport infrastructure and services.

The development of urban transport systems requires policies that involve the municipalities. The effectiveness of the proposed solutions, on the other hand, is largely determined by the reaction of the users themselves and the adaptation of the system to their expectations. Therefore, the effectiveness of a sustainable transport policy depends on the cooperation of various interest groups in the joint development of policy tools. The cooperation of so-called stakeholders at different stages, from planning to the implementation and operation of solutions, is currently considered one of the critical factors determining the success of a sustainable transport policy [2,3]. An important research gap we have noticed in the literature is the lack of an approach linking stakeholder attributes to the tasks in which they should be involved for sustainable flows in the city. In this context, the proper identification of stakeholders and the roles they play in influencing sustainable urban flows can significantly contribute to improving the functioning of a city's transport system. In doing so, it is also necessary to draw on the existing experiences of the different stakeholder groups, as well as on the scientific output in the field of urban flows of people and goods and sustainability theory. On this basis, we formulated three research questions:

1. What are the characteristics of stakeholders in a sustainable urban movement ecosystem?
2. What is the role of stakeholders in the sustainable urban passenger transport ecosystem?
3. What tasks should stakeholders in the sustainable urban passenger transport ecosystem be involved in?

Referring to the formulated objective and the adopted research questions, in the Theoretical Background we present the theoretical background on sustainable urban passenger transport and identified the theoretical and cognitive gaps on urban passenger transport stakeholders. Then, building on the methodology developed for multimodal transport [4], we developed an original research methodology on stakeholder roles in urban logistics. Furthermore, we supplemented the original methodology with an indication of how to proceed in assigning stakeholders to tasks in shaping sustainable flows of people in the city. The results of our research were focused primarily on the analysis of urban logistics stakeholders in terms of their impact on sustainable flows of people in the city. Using expert research, we assessed the level of interest of each stakeholder in the sustainable development of people flows in the city, the strength and direction of influence on such solutions in the city and the predictability of stakeholder behaviour. Subsequently, we developed stakeholder maps, based on which we assigned roles to each stakeholder group. In the Discussion, we present the combined identified stakeholder roles with recommendations for local governments in the area of involving stakeholders in tasks aimed at sustainable urban movement.

## 2. Theoretical Background

### 2.1. Sustainable Passenger Transport in the City

Cities are complex systems of relationships that are subjected to multi-directional research analyses of varying degrees of complexity [5]. One of the areas extensively discussed in the literature is the flow of people in the city, which constitutes the main subsystem of urban logistics [6,7]. Thus, the organisation of urban transport of people becomes a fundamental task, which stems from the need to enable people to move, regardless of the reasons that trigger the need as well as the way in which space is covered [8]. This is particularly important given that urban transport systems determine the quality of life in a city and influence the overall level of life satisfaction, resulting in increasing attention to them in developed economies [9–11].

Within urban passenger transport, individual and collective transport can be separated [12]. Individual transportation is contrasted with collective public transport and is characterised by the specific terms of communication and the lack of regularity. Individual transport includes transport means such as a car, a bike and a motorbike [13]. Collective transport in the city is also referred to as passenger transport [12], but also public passenger transport or public transport [14]. Collective transport is a "regular transport performed at the request of local government transport organiser only within a single municipality, two

or more municipalities, by agreement among the municipalities forming the communal interrelationship" [13]. It is implemented using public transport modes such as bus, tram, metro or light rail [15]. In addition to the two categories mentioned, Cichosz [14] mentions group transport, which currently does not have a very large share in urban transport, but exists and is under development (e.g., taxi, carpooling). The most basic way of getting around in cities, preferred by many people, is individual transport carried out primarily by car [15]. Studies show that as cities become wealthier, the number of owned and used cars tends to increase rapidly, especially in developing countries [16,17]. There are a number of complex political, economic, urban, cultural, social and psychological factors that explain this change [18–21]. Car-oriented urban development contributes to extremely serious environmental and health problems, including $CO_2$, air pollution, noise, climate change, traffic accidents and sedentary lifestyles with associated health problems. It also increases so-called social exclusion and road congestion which negatively affects local and national economies [13,22,23]. Thus, individual motorised transport poses a serious threat to both the smooth functioning of the city and the smooth movement of people as well as the economy and the environment. Measures to bring about changes in the habits, preferences and mentality of the population are therefore becoming important. They should result in an increasing interest in public transport and active mobility on foot or by bike. Thus, the main task of the city authorities is to organise public transport and other forms of movement together with their promotion, in order to reduce the use of individual (car) transport in the city [12,23].

The objectives of reducing car-based individual transport in cities are a result of the negative impacts it generates and the desire of cities for so-called sustainable development. One of its definitions is "improvement in the quality of human life within the carrying capacity of supporting ecosystem" [24]. As recorded in the Brutland Report, sustainable development is "to provide the needs of the present without compromising the needs of the future" [25]. It takes place when economic, social and environmental conditions lead to improved human activity and wellbeing without compromising the ability of future generations to experience the same [22]. Taking into account the idea of sustainability, there is also talk of the need for so-called sustainable mobility, which includes both passenger and freight flows in a city. Although research on environmentally sustainable transport started more than 20 years ago, the implementation and management of sustainable mobility still remains a major challenge worldwide [26]. As mentioned by Holden et al. [27], there is no consensus on a clear and unambiguous definition of sustainable urban passenger transport. More importantly, the changes in definition are directed towards including a broader view of the impact of transport on the environment. Holden et al. [27] highlight that researchers, in interpreting the concept, focus on the environmental impact of transport as well as the idea of social equity, the impact of transport on health and safety, the impact on urban quality of life and even the impact on economic growth. One view presents sustainable urban passenger transport as collective transport that in an ongoing way meets personal travel needs and facilitates strong communities; supports economic development and equitable social participation; promotes environmental health; and has appropriate institutional arrangements and stakeholder involvement (including sufficient sustainable funding) to deliver [28]. Ongoing research on sustainable mobility highlights the need for both technological and institutional change to achieve a radical reconfiguration of the urban transport system. Three main approaches to supporting sustainable transport can be identified [29]:

- Improving the efficiency of transport and reducing the environmental impact of vehicles (through technological improvements to existing vehicles or the development of alternative propulsion sources);
- Use of more sustainable modes of transport (increased use of public transport, walking, cycling, car sharing);
- Reducing the need to travel (urban planning, changing lifestyles, increased use of ICT, e.g., remote working).

An important aspect of sustainable cities is the creation of transport policies that aim to reduce the amount of individual transport carried out by cars. Instead, the use of public transport and other alternatives to individual transport should be promoted [22,30]. The transport of people in the city should also prevent and even effectively counteract social exclusion. This problem mainly affects the elderly, disabled, poor or those living in rural areas [31–37]. Social exclusion is defined as the process in which individuals or entire communities of people are systematically blocked from (or denied full access to) various rights, opportunities and resources that are normally available to members of a different group, and which are fundamental to social integration within that particular group [38]. This issue is of particular relevance today, when the phenomenon of an ageing population is highlighted in many countries. Public transport is seen as one that caters for the elderly and other socially excluded people. It allows them to maintain their independence and thus prevent exclusion. Unfortunately, the problem of an ageing population and its impact on the organisation of a city's passenger transport system is often ignored by government officials and decision makers. These problems are often not incorporated into transport planning and transport policies built [35,36].

An important aspect in the development of sustainable passenger mobility is the image of public transport related to the level of service quality of the transport provided. Customer satisfaction with public transport services influences increased use and consequently a reduction in car transport [39]. Studies show that public transport users who present positive opinions about public transport are more likely to show loyalty to this form of transport and are more willing to act as its ambassador [40–42]. One of the image-building factors of urban public transport providers can be the issue of environmental friendliness. Their undertaking of activities directed towards being green is beneficial for the environment, but can also bring business benefits. Companies that can demonstrate their commitment to the environment have the opportunity to gain a stronger reputation and attract customers for whom the issue of environmental sustainability is very important [30]. Thus, it is important to include sustainability goals in the strategies of transport companies and to effectively communicate their environmental commitment to customers, as more and more of them expect and appreciate such actions. Passengers are increasingly environmentally conscious, making public transport companies' commitment to sustainability a factor in building passenger loyalty to this form of transport. Thus, "green marketing" should become an integral part of public transport organisers' strategies [30]. This fact indicates that the application of an ecosystem approach offers the possibility to build an urban passenger transport system that meets certain requirements of reliability, safety, efficiency and environmental friendliness, as well as focusing on the needs of each participant [9].

Collective urban transport is identified in the literature as the main way to move towards sustainable mobility and, consequently, sustainable cities. However, it should be noted that recent years have brought a number of problems that this transport has had to face due to the COVID-19 pandemic [43–53]. The spread of the pandemic affected the lower mobility of people in cities as a result of the fear of contagion and the numerous movement restrictions introduced. This fact has contributed to positive effects in the form of lower GHG emissions [51] or less congestion. However, the pandemic has also been a series of disadvantages that have led to public transport being seen as one of the most problematic sectors. This resulted in a significant regression of collective transport. Bus stations, as well as public transport vehicles themselves due to the presence of concentrations of people in them, have been identified as high risk sites for the spread of the virus. The introduction of a number of restrictions in the operation of passenger transport aimed at reducing the risk of contagion (e.g., the abolition of additional ticket sales, the abolition of opening doors on demand, changes in timetables) has been highly disruptive to passengers [43,46,53]. These indicated factors led to a loss of confidence in public transport as a safe, reliable and convenient form of transport. Consequently, the number of passengers on public transport decreased during the pandemic. This was largely due to the choice of an alternative form of transport such as individual transport, which was considered a safer mode of travel.

Thus, individual transport, which was already strongly developed in cities, using cars, benefited [45–52]. It should also be emphasised that by not having passenger transport that is safe for people's health, life in the city is paralysed and it is then difficult to talk about sustainability. It is now important to restore confidence in public transport as a safe form of travel, which is crucial for the development of sustainable urban mobility. It is necessary to follow hygienic rules to ensure the safety of transport and consequently regain lost customers [43–46]. The time has come to think about a sustainable public transport system taking into account the new conditions and the new rules that will be dictated by the global epidemiological situation. The development of individual transport using so-called micromobility (bike sharing, scooter sharing, moped sharing) has become an important direction of action in cities (also in terms of pandemics). With it, cities have the opportunity to solve three problems at the same time: they will relieve the burden on public transport to improve its safety in terms of virus transmission, reduce road congestion with individual car transport and reduce air pollutant emissions. It should be noted that during the pandemic period, micromobility was indicated by residents as a safer form of transport than public transport. This is evident in studies showing an increase in micromobility during the pandemic (e.g., an increase in bicycle use of 187% in Beijing and an increase of 67% in New York). This situation has influenced the managers of many cities, who have made a number of decisions and measures geared towards the development of this form of transport and the reduction of road transport (construction of cycle lanes, closure of streets for car transport, financial incentives to switch to bicycles, priority for pedestrians, dedicated lanes for public transport) [43]. Micromobility is an innovative solution for urban passenger transport. It aims to provide short-distance mobility options, with a focus on the first and last mile [54]. The attractiveness of this form of transport is that it provides a flexible, sustainable, cost-effective and on-demand alternative to car transport [55]. The use of micromobility is fully in line with the current philosophy of shared mobility, which is part of the broader concept of the sharing economy [54]. Shared mobility is based on the joint use of different modes of transport. In addition to the aforementioned micromobility, another example of its application in the city is carsharing or carpooling, which also have the effect of reducing the car transport performed and thus contribute to building sustainable urban mobility [7].

The construction of a sustainable urban passenger transport system Is increasingly being based on a range of innovative ICT-based solutions. These contribute to the creation of so-called smart mobility, which is part of the smart city concept [26,56]. Smart mobility means modern transport and logistics systems using ICT to enhance their integrity with the environment and enable people and goods to move around in a safe, user and environmentally friendly manner as well as in an efficient and cost-effective way [57]. It aims to improve movement while reducing the environmental and social impacts of transport, manage congestion, reduce independent travel, encourage modal shift, reduce journey lengths and increase the efficiency of the transport system [58]. It covers many issues, such as the monitoring of behavioural aspects and mobility preferences of citizens using information and communication technology (ICT) tools. Electric vehicles will also be a vital element in future smart urban transport systems. Therefore, various electromobility-related topics can be included in smart mobility, such as EV charging and energy management with the use of smart grids, or strategies for vehicle electrification [26]. One of the solutions strongly supporting the development of smart urban mobility is ITS (intelligent transport systems), which include modern technological and organisational solutions in transport. They are among the most effective instruments for improving the efficiency and quality of a city's transport system. They allow, among other things, traffic control, the creation of special restricted traffic zones and low carbon dioxide emissions by reducing the number of private cars in city centres [59]. As research shows, the use of ITS technology can contribute to improving the safety and efficiency of the transport system and reducing environmental impacts [60]. It is undoubtedly an opportunity in the sustainable development of cities and regions [59].

Striving for sustainable urban passenger transport requires city managers to make a number of decisions and actions aimed at reducing individual car transport and developing alternative modes of travel. Other measures implemented in cities (not mentioned above) include the use of environmentally friendly modes and means of transport (e.g., trams), investing in energy-efficient means of transport, integrating the public transport system in the city, creating interchanges, building a system of Park and Ride, Bike and Ride, the use of connection search engines that will allow efficient planning of public transport journeys and the introduction of tariff-ticket integration, including ticketing in the form of an electronic city card [61], the creation of preferences for public transport, the introduction of fees for entering city centres, the restriction (or closure) of car traffic in city centres, the introduction of small buses into public transport, moving at a higher frequency than buses and the use of various telematic solutions [62].

In practice, there are many solutions for building a sustainable transport system for people in a city. Many of the decisions to implement these solutions are made by local authorities in cities. They are faced with the need to create an environment that is conducive to the initiation, development and implementation of solutions that improve the mobility of all traffic participants and improve the functioning of the whole city [59]. In this respect, however, it is important for local authorities to cooperate with different stakeholders [56,62]. Additionally, Le Pira et al. [63] emphasise the need to involve a wide range of stakeholders in transport planning. Among them, they mention citizens, policy makers, public institutions, local communities, governmental organisations, NGOs, public transport operators, experts, retailers, the private sectors and the third sector. Taniguchi [64] points out that collaboration between stakeholders and local authorities is crucial for the successful implementation of urban logistics initiatives, including in the area of transport. He points out the inclusion of stakeholders in the planning, implementation and evaluation of proposed urban policies. The diversity of urban logistics stakeholders requires that their needs be known, understood and taken into account in the solutions to be implemented. At the same time, the success or failure of urban efforts is often linked to the level of stakeholder involvement in the design, planning and implementation of solutions [65]. Rubini and Lucia [66] highlight the need for stakeholder engagement as a strategic factor in any decision-making process. Urban logistics stakeholders represent two groups: public stakeholders (mainly local and regional authorities and public companies, e.g., public transport operators in the city) and private stakeholders (a very diverse group including, for example, residents, businesses, but also tourists or other visitors to the city). It is the local authority that has the greatest interest in implementing an effective and efficient transport system, mainly because it is responsible for improving the quality of life in the city and for planning, organising and controlling the different logistical areas and initiatives in urban logistics. Public transport operators mainly focus on providing the best transport services for its users. Residents, on the other hand, are the group most affected by the transport nuisance associated with the external costs generated. Thus, they are the group most interested in a sustainable urban transport system [67]. Taniguchi [64], in his analyses, points out the importance of stakeholder collaboration to obtain reliable data, which is essential to understand the current state of the problems involved and, consequently, to find innovative solutions. He emphasises that the exchange of data between private and public sector stakeholders is beneficial for the planning of sustainable transport in a city. The introduction by city authorities of policies aimed at developing sustainable passenger transport requires public support for the successful implementation of the proposed solutions. This is all the more so as these policies often imply significant changes in the behaviour of participants in the transport system. Achieving such changes requires public acceptance. Thus, it is necessary to actively involve stakeholders representing the wider public in the development of transport policies, with the aim of leading to better and more acceptable decisions [29]. In conclusion, it is important to emphasise the important role that different stakeholder groups play in shaping sustainable urban

passenger transport. This should be reflected in the way stakeholders are included in the development and implementation of urban transport policies.

### 2.2. *Theory and Cognitive Gaps in the Area of Passenger Transport Stakeholders in the City*

The issue of passenger transport forms part of one of the more important strands of urban logistics. This is evidenced by the increasing number of publications in this field in recent years. This is undoubtedly due to the spatial expansion of cities, the increase in population density in cities, the increasing number of institutions and companies in a city. This, in turn, causes communication problems because of the need for movement. Due to the significant importance of the issue in the literature research, we took on the challenge of presenting an up-to-date and comprehensive bibliometric analysis to describe and evaluate the scientific literature on passenger transport. The analysis was based on two bibliographic sources—the Web of Science database and Scopus. The study included publications with the following terms in their titles and topics: passenger transport in the city, passenger transport, public transport, public sustainability transport, movement of people in the city, stakeholders of passenger transport in the city. The VOSviewer ver. 1.6.18 tool was used for bibliometric analysis, which allowed us to formulate conclusions affecting the validity of the research we undertook.

In step I of the research, we conducted a bibliographic review of the databases with a view to their applicability for creating a numerical listing of publications. In step II, we selected a suitable database containing literature items from the research area. Step III of the research consisted of identifying key words, which, together with the results, are presented in Table 1. Step IV, on the other hand, consisted of conducting a web mapping with the VOSviewer tool, on the basis of which we analysed the obtained results.

**Table 1.** Results of bibliometric analysis.

| Query Wording (QW) | Web of Science | | Scopus | |
| --- | --- | --- | --- | --- |
| | QW Appearance as Topic of Publication | QW Appearance in Title of Publication | QW Appearance as Topic of Publication | QW Appearance in Title of Publication |
| Passenger transport in the city | 2067 | 8 | 5159 | 19 |
| Passenger transport | 12,311 | 1190 | 20,425 | 1659 |
| Public transport | 33,436 | 4684 | 56,467 | 6418 |
| Public sustainability transport | 1882 | 45 | 6114 | 151 |
| Movement of people in the city | 2633 | 8 | 3908 | 11 |
| Stakeholders of passenger transport in the city | 80 | - | 98 | - |

Source: own study.

As the table above shows, the topic of public transport is the most popular among researchers in the selected area. In the Web of Science database, more than 33,000 literature items can be found on this topic, while in the Scopus database, more than 56,000 items can be found. The area of passenger transport was in second place. The phrase most frequently appearing in the title of publications was also public transport. However, the entire phrases "Passenger transportation in the city" and "Movement of people in the city" were only found 8 times in the title of publications according to the Web of Science database and 19 and 11 times, respectively, according to the Scopus database. By far the lowest popularity among researchers is for topics involving stakeholders in passenger transport in the city. The analysis showed that no publication title covered this topic. In contrast, only 80 publications were found in the Web of Science database that address the issue of urban passenger transport stakeholders in their content. In the Scopus database, only 98 such publications were identified. Thus, a very low level of research in the field of urban passenger transport stakeholders is evident.

For the most frequently addressed public transport issue in the literature and the least frequently addressed passenger transport stakeholder issue, using the VOSviewer tool, we developed maps of research trends undertaken in publications (Figures 1 and 2). Due to the larger number of library resources in the Scopus database, we first analysed the most frequently addressed public transport issue in the literature based on the co-occurrence of keywords provided by authors from the collection of publications from 1980–2022 (Figure 1).

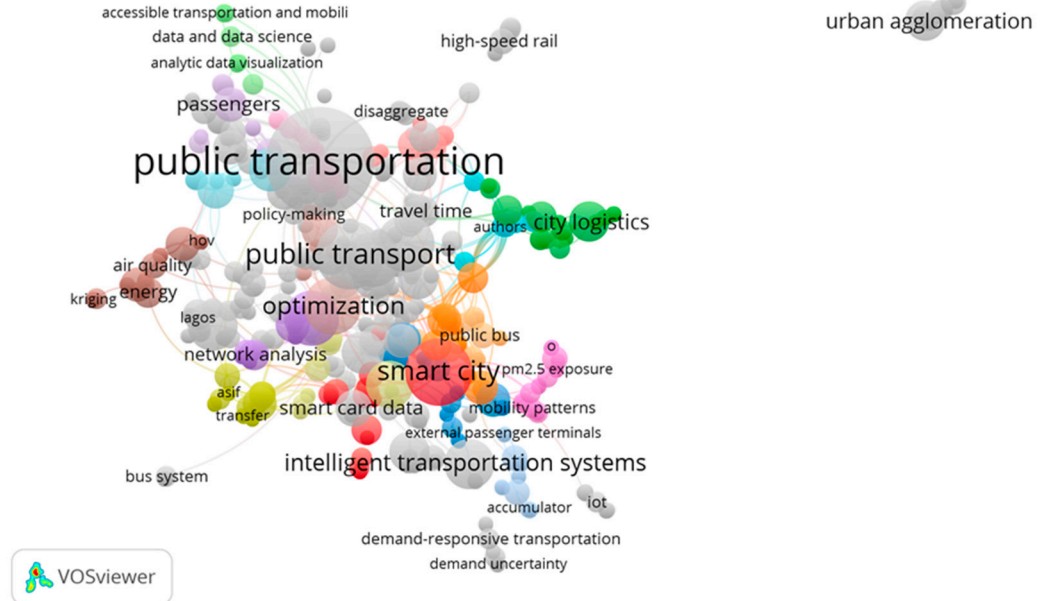

**Figure 1.** Map of research trends based on co-occurrence of author keywords in public transport publications from the Scopus database between 1980 and 2022. Source: own work using VOSviewer.

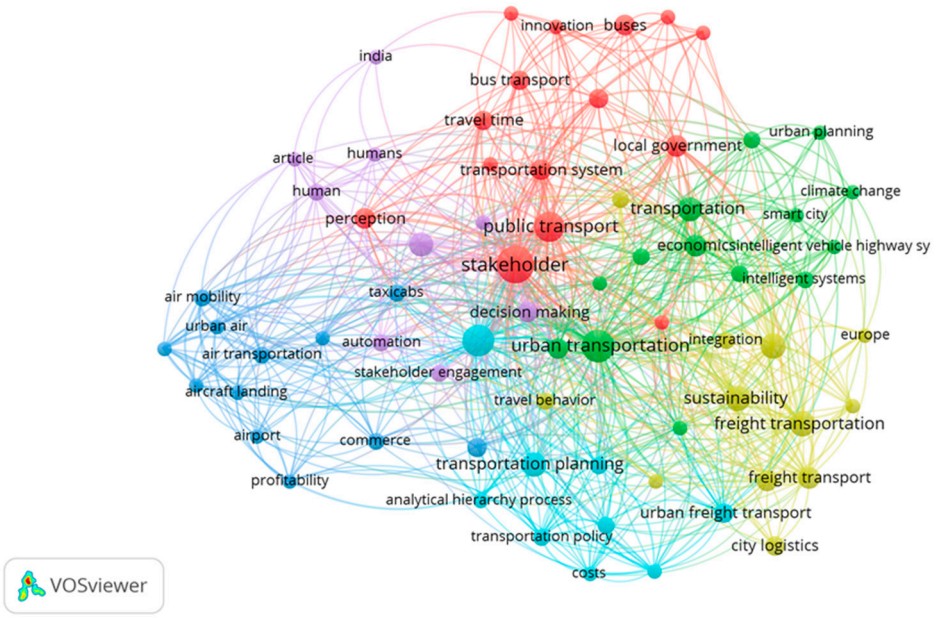

**Figure 2.** Map of research trends based on the co-occurrence of the authors' keywords in publications on the stakeholders of passenger transport from the Scopus database in 2005–2022. Source: own work using VOSviewer.

Through the analysis of the map in Figure 1, we have noticed a strong trend of linking research relating to public transport with optimisation and the context of intelligent transport systems and smart cities. It should be noted that this may be due, among other

things, to the fact that, to a large extent, modern solutions in public transport are related to advanced technologies and their optimisation. Analyses relating to city logistics have also proved to be a strong research trend. As a context imposed on many analyses from different areas, this trend reflects a change in the perception of public transport from a narrow focus on technology alone, to a process encompassing many aspects, not only technological. Interesting, discernible trends were also those that appeared on the periphery of the map, such as urban agglomeration and the bus system.

From the perspective of the search for research gaps and new research topics in the field of urban logistics, areas that appear less frequently in the set of papers considered in the bibliometric analysis, but that co-occur with areas that appeared frequently in the bibliometric analysis (such as accessible transport and mobile smart and sustainable transport) are also of interest. The analysis covers the period from 2005 to 2022 due to the first papers in the subject area, retrieved in 2005 (Figure 2).

Analysing the research trend map in Figure 2, it is possible to see that there is no dominant trend centred around the area concerning the stakeholders of passenger transport in the city. This demonstrates the multifaceted and multidimensional nature of the issue. However, it can be considered that there is an overlap between the topics covered in the area of urban transportation and the area of public transport. This is evident firstly due to the proximity of the location of these two formulations, but also the size of the indication (diameter of the points) of such trend, resulting from the number of articles published in this area. It should also be noted that, compared to the analysis of urban transport, there is a lack of clearly marked research clusters focused on the issue of passenger transport stakeholders. There are more studies in the literature that cover urban freight transport issues. Such a trend demonstrates the great potential of the research we have undertaken.

Fitting into the theory and cognitive gap regarding the stakeholders of urban passenger transport, we took a broader look at the problem and started our research with the identification of urban logistics stakeholders. More literature can be found in this area. Stakeholder groups are covered very differently in them. Katsela and Browne [68], elaborating on the research on stakeholder identification previously conducted by Taniguchi et al. [69] and Benjelloun et al. [70] point to the five most commonly mentioned stakeholder groups in the literature: shippers, freight carriers, administrators, residents and others. The last category may include non-governmental organisations (NGOs) and property owners [71]. By contrast, combining passenger and freight transport, Kiba-Janiak [72] points to public organisations (representatives of the local government including the decision making and controlling bodies, the executive bodies and the city council as a support apparatus including, planning, infrastructure, transport, IT and telematics, development and promotion departments, companies offering public transport on behalf of the city council and municipal companies), private organisations (companies/people offering public transport on behalf of the city council, municipal companies, etc.), private organisations (freight forwarders/receivers, transport companies, freight carriers, logistics companies, couriers, private companies offering public transport, manufacturing companies), non-governmental organisations (associations, foundations), the public: unorganised (residents, consumers, visitors to the city) and organised (e.g., grassroots movements). Looking even more broadly at the problem of stakeholder groups, Dohn et al. [73] list 13 stakeholder groups: inhabitants (I), production enterprises (PE), trade and service enterprises (TSE), transport and logistics enterprises (TLE), environmental organisations (EO), organisations related to health care (OHC), organisations related to arts and culture (OAC), organisations related to public safety (OPS), organisations related to sport and recreation (OSR), R&D organisations (R+D), municipal management enterprises (MME), organisations working for the benefit of disabled people (ODP) and other cities (OC). However, in order to analyse stakeholder roles in detail from the perspective of sustainable urban logistics, we decided to make these groups more specific. This was because, within a given group, it was still possible to identify stakeholders with different, sometimes even conflicting, interests in the development of sustainable urban logistics. In this way, we identified a group of

30 stakeholders, including inhabitants (I), production companies (PC), freight transport companies (logistics operators, shippers and carriers) (FTC), environmental organisations (EO), health-related organisations (HO), arts- and culture-related organisations (CO), public safety organisations (PSO), organisations related to sport and recreation (OSR), R&D organisations (R+D), organisations working for the benefit of disabled people (OD), other cities (OC), organisations related to food delivery (FO), companies designing smart and logistic solutions in the city (ISC), courier and postal companies (KEP), hypermarkets and discount shops (HiD), wholesale and retail chains (DN), wholesalers and retail shops (DS), service companies (SC), waste management companies (WMC), other municipal companies (OCO), tourists and visitors (TI), educational organisationsI), taxi corporations (TC), transport infrastructure managers (IM), micromobility operators (MMO), public transport organisations (OPT), estate managers (EM), media suppliers (MS), religious organisations (RO) and social welfare organisations (SO).

Stakeholder groups themselves, however, are not indicative of how to work with them. Factors that influence how to work with stakeholders are the level of interest, the direction and strength of the stakeholder's influence on the area and the level of predictability of their behaviour. Following this characterisation, a strand of research on stakeholder roles can be identified in the literature. Research focused on stakeholder roles in innovation ecosystems was conducted by Iansiti and Levinen [74], mentioning the keystone, dominator and niche player. In the following years, other authors also undertook identifying roles in innovation ecosystems. Five roles are indicated by Bosch-Sijtseemaa and Bosch [75], mentioning keystone, dominator, complementor, integrator and leader. Dedehayir et al. [76], on the other hand, identify direct value, creators, value creator, supporters and entrepreneurs. Stakeholder roles in relation to logistics have been studied extremely rarely. There are no literature references on stakeholder roles in urban logistics. Kramarz et al. [4] investigated stakeholder roles in a cross-border multimodal transport ecosystem. The combination of logistics aspects with their regional impacts is most consistent with the urban logistics research area. Therefore, we used the roles identified by Kramarz et al. [4] and related them to urban logistics stakeholders (Table 2).

**Table 2.** A set of stakeholder roles in the city logistics ecosystem.

| Role No. | Role of the Stakeholder | Characteristics |
|:---:|:---:|:---|
| 1 | Development leader | High predictability, high positive power of influence and high interest in improving sustainable flows in the city indicate that the stakeholder is key and will strive to develop the ecosystem. At the same time, high power of influence and a high degree of predictability are characteristics that indicate that the stakeholder has influence over other stakeholders. |
| 2 | Unpredictable major player | The high level of interest and the high positive power of influence demonstrate the crucial importance of this actor in improving sustainable flows in the city. However, the unpredictability of its behaviour represents an increased risk, as it is more difficult to adapt logistical solutions under such uncertainty. |
| 3 | High-risk influencer | The low interest and high positive influence of this player on the development of the ecosystem indicates its importance in supporting the improvement of sustainable urban flows. However, the unpredictability of its behaviour means that this player can influence the system more or less strongly depending on the current determinants of its interest unrelated to sustainable urban flows. |
| 4 | Patron | High predictability in behaviour and high positive power of influence mean that this player will strongly support the improvement of sustainable flows in the city despite low self-interest. |

**Table 2.** *Cont.*

| Role No. | Role of the Stakeholder | Characteristics |
|---|---|---|
| 5 | Beneficiary | The high interest and high predictability of this player, while having a low positive power of influence, means that he is a recipient of solutions for improving sustainable flows in the city. The effects of ecosystem development are important to him, however, he himself has little influence on ecosystem development. |
| 6 | Unpredictable position | The participant's high interest in improving sustainable flows in the city and, at the same time, low positive power of influence and low predictability mean that he will benefit from improving sustainable flows in the city. His behaviour is unpredictable which means that he may change his actions directed towards ecosystem development depending on his current interests. |
| 7 | Unaware | A participant who has low positive power of influence, low interest and at the same time is unpredictable in his behaviour does not play an important role in improving sustainable flows in the city. He is unpredictable but manageable. |
| 8 | Neutral | High predictability, low positive power of influence and low interest result in a participant that is not a threat, but at the same time has no influence on sustainable flow solutions in the city. |
| 9 | Resistant but predictable | High predictability, low negative power of influence and low interest mean that a participant can interfere in a minor way with the development of sustainable flows in the city. |
| 10 | Unpredictable opponent | Low predictability, low negative power of influence and low interest mean that a participant can interfere in a minor way with the improvement of sustainable flows in the city. |
| 11 | Declared opponent | Participant strongly inhibiting the improvement of sustainable flows in the city. Its highly predictable negative impact on the ecosystem, with a concomitant low interest in improving sustainable flows in the city, make it a major threat to ecosystem development. |
| 12 | Unaware unpredictable beneficiary | The unpredictable behaviour of this participant and the insignificant inhibitory power to improve sustainable flows in the city counteract the high interest of this participant. The participant is unaware of the benefits of ecosystem development and is therefore an uninformed recipient of solutions who takes inhibitory actions without being aware of their negative effect on his own interests. |
| 13 | Unaware beneficiary | The low inhibitory power of influence, which is contrasted with a high level of interest and high predictability, indicates that the participant is not aware of the benefits to him from the development of the ecosystem. |
| 14 | Sceptic | The high, inhibiting power of influence, low level of interest and low predictability indicate that the participant can strongly interfere with the improvement of sustainable flows in the city. It can be influenced mainly by administrative instruments. |
| 15 | Unpredictable key player | High negative power of influence, high self-interest and low predictability are the characteristics of a participant who has difficulty in adapting his business model to the challenges of sustainable urban flows. Such a participant has a predisposition to unconsciously disrupt the activities of other participants due to the fact that the other stakeholders are highly dependent on it. |
| 16 | Inhibiting key player | High negative power of influence, high interest and high predictability are characteristics that point to a crucial but inhibiting role for this stakeholder in improving sustainable flows in the city. He may be unaware of high interest or the stakeholder may be struggling to adapt its business model to the challenges faced by ecosystem participants. Its high predictability provides an opportunity for targeted action to change the direction of the power of influence. |

Source: own study based on [4].

### 3. Methodology

The research on stakeholder roles follows the methodology initiated by Kramarz et al. [4] for the multimodal transport ecosystem. The implementation of this methodology for the study of stakeholder roles in urban logistics results in a 4-step research process (Figure 3) aimed at indicating how the city should involve individual stakeholders, by virtue of their roles, in improving sustainable flows of people in the city. The same research steps are currently being carried out for the sustainable freight transport sub-system in the city and the waste management sub-system in the city.

| | Research stage | Literature research | Empirical studies |
|---|---|---|---|
| STAGE 1 | Identification of challenges in the area of sustainable urban passenger transport and urban logistics stakeholders | 1. Conduct a literature review on sustainable urban flows of people 2. Conduct a systematic literature review using the VOSviewer tool to identify theoretical and cognitive gaps in the area of passenger transport stakeholders | |
| STAGE 2 | Stakeholder analysis of the urban logistics ecosystem | Choice of stakeholder analysis methodology | 1. Conducting expert studies (assessing strength and direction of impact, level of predictability and interest) 2. Development of stakeholder maps |
| STAGE 3 | Identification of stakeholder roles in urban logistics | Interpretation of stakeholder roles | 1. Assigning roles to stakeholders in sustainable people flows in Polish cities |
| STAGE 4 | Development of recommendation for local governments in the field of cooperation with stakeholders in the area of sustainable flows | | 1. Conducting expert panel research – guidelines for assigning roles to tasks. 2. Assigning roles of stakeholders to tasks related to shaping sustainable flows of people in the city |

**Figure 3.** Research methodology. Source: own study.

Phase one of the research concerns the selection of urban logistics stakeholder groups in terms of their impact on sustainable urban flows. The research considered the group of 13 urban logistics stakeholders identified by Dohn et al. [73], detailed to 30 sustainable urban logistics stakeholders. In this phase, a literature review was conducted focusing attention on two aspects: the challenges in the area of sustainability-oriented urban passenger transport and the stakeholders of urban logistics, together with the identification of a theoretical and cognitive gap in the area of urban passenger transport stakeholders. The results obtained along with the set of identified urban logistics stakeholders were used in the next two stages of the research, in which expert research was applied. We used a deliberate selection of experts, and thus we obtained the most competent people for the study. For this purpose, we used the method of recommendation (nomination or co-nomination), by identifying persons widely recognised as an authority or expert in the field of urban logistics in the broad sense, taking into account both their involvement in project work in this area, but also guided by an analysis of publications in this field. In addition, we took into account experts from outside the scientific community, representing representatives of local authorities and business representatives. The group of experts was unchanged in both stages of the research and consisted of 15 experts. Due to the research's focus on Polish cities, the experts included only Poles. In the research, we used a research questionnaire. After receiving the feedback, we conducted a statistical analysis to verify the differences in expert opinions: we checked the normality of the distributions of expert opinions and the equality of their variances. We calculated the correlation coefficients between the evaluations of the individual experts and then assessed their level. Due to the high correlation coefficients obtained, we considered that agreement among the experts had been reached, and therefore decided that there was no need for another round of testing.

Stage two of the research concerned the characterisation of urban logistics stakeholders in the area of shaping sustainable flows of people. For this purpose, the experts assessed the level of interest of each stakeholder in the sustainable development of people flows in the city, the strength and direction of influence on such solutions in the city and the predictability of the stakeholder's behaviour. The strength of influence was evaluated on a scale of (0–5), where 0 means no influence and 5 means high influence. A similar scale was used to assess predictability and level of interest. The direction of influence could be positive and negative, where a negative direction means a negative influence on the development of sustainable urban passenger transport and a positive direction means a positive influence on the development of sustainable urban passenger transport. Based on the averaged results of the experts' opinions, two stakeholder maps were developed. The first map classifies stakeholders according to the criterion of strength and direction of influence and predictability, while the second map classifies stakeholders according to the criterion of strength and direction of influence and level of interest. The location of a stakeholder in one of the quadrants of the first map and in one of the quadrants of the second map corresponds to the role of the stakeholder in the ecosystem of sustainable urban passenger transport. This is already the third stage of the research, which required the introduction of stakeholder roles through a review of the literature research in this field and then an analysis of the results obtained in stage two included in the stakeholder maps. In accordance with the characteristics thus presented, each stakeholder was assigned to one of sixteen potential roles (Table 2). The research carried out in Polish cities allowed the stakeholders to assign roles in the ecosystem of sustainable urban passenger transport. The fourth stage links the identified stakeholder roles with recommendations for local governments in the area of involving stakeholders in tasks aimed at sustainable urban passenger transport. In this stage, experts assigned stakeholder characteristics (direction and strength of influence, level of interest, predictability) to individual tasks. Thus, each task was given a set of stakeholder characteristics for its implementation. On this basis, the tasks recommended for each stakeholder role were indicated (Figure 4).

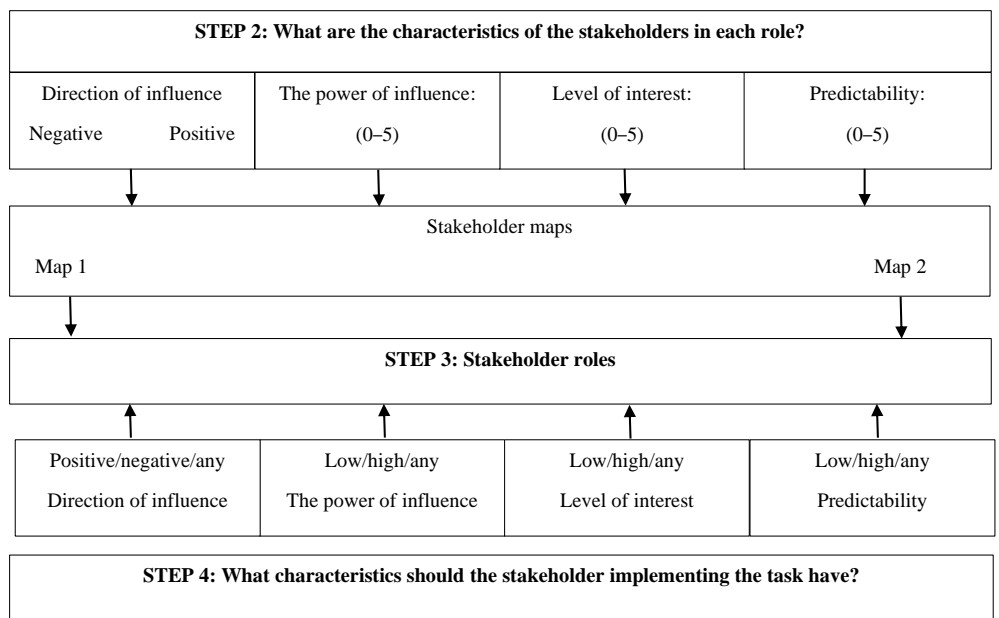

**Figure 4.** Links between the research phases 2–4. Source: own study.

## 4. Results

The stakeholder groups identified in Section 2.2 were subjected to expert evaluation in terms of the strength and direction of the stakeholder's influence on the development of sustainable passenger transport in the city, the level of interest the stakeholders have in this development and the predictability of their behaviour in the context of sustainable passenger transport. The resulting data made it possible to prepare two stakeholder maps. The first takes into account the criteria: strength and direction of influence and predictability of stakeholder behaviour (Figure 5). The second takes into account the criteria: strength and direction of influence and level of stakeholder interest (Figure 6).

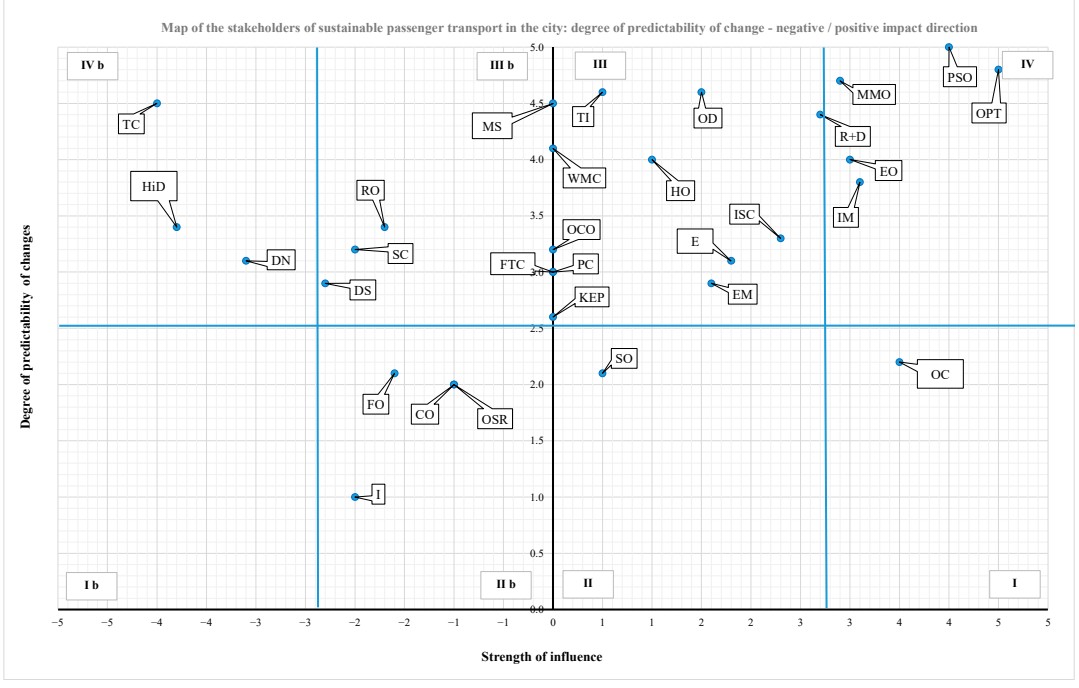

**Figure 5.** First stakeholder map: strength and direction of influence—degree of predictability. Source: own study.

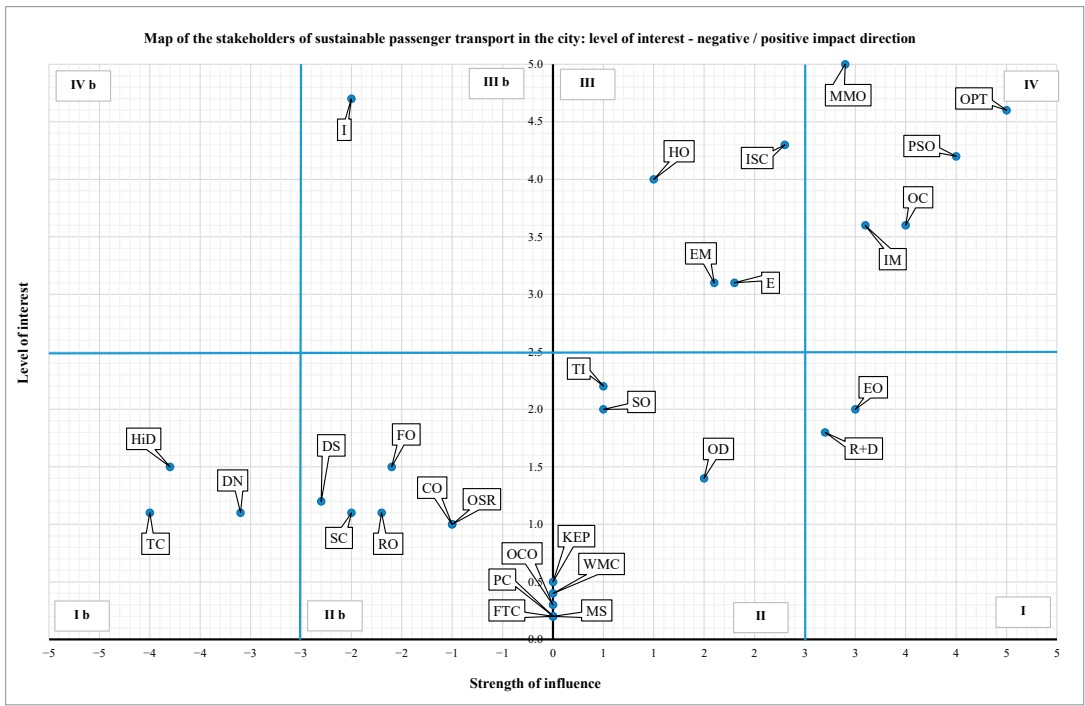

**Figure 6.** Second stakeholder map: strength and direction of influence—level of interest. Source: own study.

The map analysis shows that of the 30 urban logistics stakeholders analysed, 24 were identified as stakeholders in the city's passenger transport sub-system. Six stakeholders were identified by experts as having no influence on the development of sustainable passenger transport in the city (they do not inhibit or support its development). At the same time, all of these stakeholders are characterised by a lack of interest in the development of sustainable passenger transport in the city (scores ranging from 0.2–0.5) and a high predictability of their behaviour in relation to this research area (scores above 2.5). These are production companies (PC), courier and postal companies (KEP), freight transport companies (FTC), media suppliers (MS), waste management companies (WMC) and other municipal companies (OCO). Taking into account the results of the expert assessment, these six stakeholders were not included in the further research related to the attribution of roles in the development of sustainable passenger transport policy. These are stakeholders who will not influence this policy in any way, have no interest in this transport and their predictability allows us to conclude that this area will not change. Thus, further research was focused on a group of 24 stakeholders in sustainable passenger transport in the city.

When analysing the strength of influence, it is apparent that 14 stakeholders (out of 24) were assessed as supporting the sustainable development of passenger transport in the city. In contrast, 10 are seen as inhibiting this development. Within this group, three stakeholders have a high, inhibiting power of influence. These are hypermarkets and discounts shops (HiD), wholesale and retail chains (DN) and taxi corporations (TC). At the same time, these groups are predictable in their behaviour and have a low interest in the development of sustainable passenger transport in the city. Among the stakeholders inhibiting development, attention should also be paid to residents who are the main participants in the urban passenger transport sub-system. Residents have been identified as an inhibitor to the development of sustainable transport. However, their power is relatively low. The unpredictability of residents in terms of their behaviour is furthermore a problematic issue. This may have to do with the different measures that are taken in the city in the area of sustainability of people flows. Some of them are quite readily accepted by residents (e.g., micromobility), while others (e.g., restrictions on car traffic) may arouse opposition. At this point, it is also worth pointing out a contradiction that exists for

residents. As indicated by the expert assessments obtained, they are the group that hinders the development of passenger transport towards sustainability, while at the same time having a very high level of interest in this development (4.7). In addition to residents, six more stakeholders inhibit the development of sustainable passenger transport in the city to a low degree (rating below −2.5). It can be assumed that their negative influence is linked to the low potential benefits that all these groups can gain from sustainable transport (level of interest below 2.5). Against this background, the distinctiveness of residents, whose benefits have been identified at a high level, is even more pronounced. This aspect should be skillfully exploited by city authorities when shaping transport policy. Among the 14 stakeholders supporting the development of sustainable passenger transport, half are those with a high strength of influence (above 2.5). Within this group, the highest influence was indicated for two stakeholders: public transport organisations (OPT), rating 4.5 and public safety organisations (PSO), rating 4. Other stakeholders with high influence included micromobility operators (MMO), transport infrastructure managers (IM), other cities (OC), R&D organisations (R + D) and environmental organisations (EO). It is positive that of the 14 stakeholders supporting the development of sustainable passenger transport, only two were rated at a low level of predictability. Thus, it can be assumed that their behaviour can be analysed and predicted relatively well. On the other hand, the low level of predictability of a stakeholder such as other cities is quite worrying, especially as it has a high level of interest in the development of sustainable passenger transport. This can generate problems for potential cooperation between different neighbouring cities. Of the 14 stakeholders with a positive influence on the development of sustainable passenger transport, nine have a high level of interest in this development. Within this group, two stakeholders are clearly visible: micromobility operators (MMOs), for whom the level of interest was rated at the maximum number of points (5), and public transport organisations (OPT), whose interest was set at 4.6.

According to the proposed research methodology, the 24 stakeholders identified were assigned roles that they should play in the development of sustainable passenger transport in the city (Table 3). As mentioned earlier, six stakeholders were not subject to this research phase due to the fact that they are stakeholders in urban logistics but not in passenger transport. Their characteristics indicated zero influence, high predictability and no interest in the development of sustainable passenger transport in the city.

As shown in Table 3, out of the total potential 16 roles, sustainable passenger transport stakeholders were assigned a total of 10 different roles. The roles that were not assigned were high-risk influencer (3), unpredictable position defender (6), unaware beneficiary (13), sceptic (14), unpredictable key player (15) and inhibiting key player (16). The ten assigned roles include: development leader (4 stakeholders), unpredictable major player (1 stakeholder), patron (2 stakeholders), beneficiary (4 stakeholders), unaware (1 stakeholder), neutral (2 stakeholders), resistant but predictable (3 stakeholders), unpredictable opponent (3 stakeholders), declared opponent (3 stakeholders) and unaware unpredictable beneficiary (1 stakeholder). In a further stage of the research, stakeholder roles were assigned to the eight identified tasks related to the creation of a sustainable passenger transport policy in the city. The guidelines indicated by the experts related to the description of the role that must be fulfilled for it to be assigned to a task were used for this. This was in line with the research steps presented in Figures 3 and 4, under Section 3. The guidelines obtained from the expert research on how to assign roles to tasks and, consequently, stakeholders to tasks, are presented in Table 4.

Taking into account the guidelines defined by the experts (Table 4), stakeholder roles were assigned to each task, which are in line with the adopted guidelines. Then, using the data in Table 3, stakeholders were assigned to the tasks to represent each role (Table 5).

**Table 3.** Stakeholder roles in the quest for sustainable urban passenger transport.

| Name of Stakeholder | Stakeholder Acronym | First Stakeholder Map (Quadrant Number) | Second Stakeholder Map (Quadrant Number) | Stakeholder Role (Role Number from Table 2) |
|---|---|---|---|---|
| Inhabitants | I | IIb | IIIb | Unaware, unpredictable beneficiary (12) |
| Environmental organisations | EO | IV | I | Patron (4) |
| Health-related organisations | HO | III | III | Beneficiary (5) |
| Arts and culture-related organisations | CO | IIb | IIb | Unpredictable opponent (10) |
| Public safety organisations | PSO | IV | IV | Development leader (1) |
| Organisations related to sport and recreation | OSR | IIb | IIb | Unpredictable opponent (10) |
| R&D organisations | R&D | IV | I | Patron (4) |
| Disability-related organisations | OD | III | II | Neutral (8) |
| Other cities | OC | I | IV | Unpredictable major player (2) |
| Organisations related to food delivery | FO | IIb | IIb | Unpredictable opponent (10) |
| Companies designing smart and logistic solutions in the city | ISC | III | III | Beneficiary (5) |
| Hypermarkets and discount shops | HiD | IVb | Ib | Declared opponent (11) |
| Wholesale and retail chains | DN | IVb | Ib | Declared opponent (11) |
| Wholesalers and retail shops | DS | IIIb | IIb | Resistant but predictable (9) |
| Service companies | SC | IIIb | IIb | Resistant but predictable (9) |
| Tourists and visitors | TI | III | II | Neutral (8) |
| Educational organisations | E | III | III | Beneficiary (5) |
| Taxi corporations | TC | IVb | Ib | Declared opponent (11) |
| Transport infrastructure managers | IM | IV | IV | Development leader (1) |
| Micromobility operators | MMO | IV | IV | Development leader (1) |
| Public transport organisations | OPT | IV | IV | Development leader (1) |
| Estate managers | EM | III | III | Beneficiary (5) |
| Religious organisations | RO | IIIb | IIb | Resistant but predictable (9) |
| Social welfare organisations | SO | II | II | Unaware (7) |

Source: own study.

**Table 4.** Guidelines for assigning stakeholder roles in a sustainable urban passenger transport policy.

| No | Task within the Framework of the Formulated Policy for Sustainable Passenger Transport in the City | Characteristics of the Stakeholder Role Required for Each Task | | | |
| --- | --- | --- | --- | --- | --- |
| | | Power of Influence | Direction of Influence | Level of Interest | Level of Predictability |
| 1 | Opinion on solutions | High and low | Positive and negative | High and low | High and low |
| 2 | Reporting needs | High and low | Positive and negative | High and low | High and low |
| 3 | Creating innovation | High and low | Positive | High | High and low |
| 4 | Investment execution | High | Positive | High | High and low |
| 5 | Involvement in the development of the city's strategy | High | Positive and negative | High and low | High |
| 6 | Organisation of public spaces | High | Positive and negative | High | High and low |
| 7 | Evaluation of effectiveness and usability of solutions | High and low | Positive and negative | High and low | High and low |
| 8 | Promotion of activities and solutions | High | Positive | High and low | High and low |

Source: own study.

**Table 5.** Assignment of stakeholders in the development of a sustainable passenger transport policy in the city.

| No | Task within the Framework of the Formulated Policy for Sustainable Passenger Transport in the City | Stakeholder Role (Role Number—Table 2) | Name of Stakeholder—Acronym |
| --- | --- | --- | --- |
| 1 | Opinion on solutions | All 16 roles (1, 2, 3, 4, 5, 6, 7, 8, 9, 10, 11, 12, 13, 14, 15, 16) | I, EO, HO, CO, PSO, OSR, R&D, OD, OC, FO, ISC, HiD, DN, DS, SC, TI, E, TC, IM, MMO, OPT, EM, RO, SO |
| 2 | Reporting needs | All 16 roles (1, 2, 3, 4, 5, 6, 7, 8, 9, 10, 11, 12, 13, 14, 15, 16) | I, EO, HO, CO, PSO, OSR, R&D, OD, OC, FO, ISC, HiD, DN, DS, SC, TI, E, TC, IM, MMO, OPT, EM, RO, SO |
| 3 | Creating innovation | Development leader (1) Unpredictable major player (2) Beneficiary (5) Unpredictable position defender (6) | PSO, IM, MMO, OPT, OC, HO, ISC, E, EM |
| 4 | Investment execution | Development leader (1) Unpredictable major player (2) | PSO, IM, MMO, OPT, OC |
| 5 | Involvement in the development of the city's strategy | Development leader (1) Patron (4) Declared opponent (11) Inhibiting key player (16) | PSO, IM, MMO, OPT, EO, R&D, HiD, DN, TC |
| 6 | Organisation of public spaces | Development leader (1) Unpredictable major player (2) Unpredictable key player (15) Inhibiting key player (16) | PSO, IM, MMO, OPT, OC |
| 7 | Evaluation of effectiveness and usability of solutions | All 16 roles (1, 2, 3, 4, 5, 6, 7, 8, 9, 10, 11, 12, 13, 14, 15, 16) | I, EO, HO, CO, PSO, OSR, R+D, OD, OC, FO, ISC, HiD, DN, DS, SC, TI, E, TC, IM, MMO, OPT, EM, RO, SO |
| 8 | Promotion of activities and solutions | Development leader (1) Unpredictable major player (2) High-risk influencers (3) Patron (4) | PSO, IM, MMO, OPT, OC, EO, R&D |

Source: own study.

In accordance with the adopted guidelines, all 24 stakeholders in sustainable urban passenger transport have been assigned tasks. Thus, each of them should be included by

the authorities of Polish cities in the shaping of a sustainable urban passenger transport policy, observing the principles in accordance with Table 5.

## 5. Discussion

The methodology we have adopted is different from the existing research on stakeholders of urban mobility [56,62–64]. Publications in the field of stakeholders of urban mobility show different stakeholder perspectives and, above all, target different research problems. Depending on the issue of stakeholder influence addressed, different groups of stakeholders are selected, as well as different research methodologies. The most common research strand in the area of stakeholders of urban mobility is the context of sustainability. In this area, Brůhová Foltýnová et al. [77] conducted a study that aimed to identify the main perspectives on the different options of sustainable urban mobility. In their research, they selected stakeholders who share the same vision and at the same time are able to effectively influence urban mobility policy. In relation to our research, they therefore focused on the groups we distinguished: leaders, unpredictable key players and patrons. To identify common viewpoints, the authors used a combination of quantitative and qualitative methods. The Q method they used is suitable for identifying groups of opinions without quantifying their relative importance in the population as a whole. This type of analysis makes it possible to work with a relatively small, purposefully selected research sample. The use of such an adopted methodology allowed the segmentation of stakeholders, active at local or national level, who share the same vision of sustainable urban mobility. Data were collected through structured interviews. Stakeholders participating in the study included those who shape the transport policy of the largest Czech cities with more than 50,000 inhabitants, as well as smaller cities that are among the leaders in sustainable urbanism in the Czech Republic. Stakeholder segmentation is a concept similar to our proposed method. However, in our research we consider all stakeholders, regardless of their vision of a sustainable urban logistics system. A similar problem was addressed in the study by Rześny-Cieplińska i Szmelter-Jarosz [78], who also oriented their research towards sustainability. The authors identified the most environmentally friendly urban logistics measures from the perspective of different stakeholders. They conducted their research in three phases: case study, Delphi interviews and statistical analysis. The authors indicate that the voices of stakeholders should be important factors taken into account in the preparation of regulations, planning and implementation of new investments and the creation of public spaces. The tasks indicated by the authors have been taken into account in our research and made more specific. We analysed the stakeholders themselves in a different way, giving them roles and making their involvement dependent on them in tasks concerning the improvement of sustainable flows of people in the city. Lindenau and Böhler-Baedeker [79] considered a similar problem, which also fits in with the focus of our paper, by investigating the involvement of citizens and other stakeholders in the sustainable development of urban passenger transport. The authors, like us, carried out their study using expert opinion, but did not take into account the strength and direction of stakeholder influence, the level of interest and their predictability. In contrast, a different perspective on the study of the city's stakeholders has been adopted by [2,80,81] by identifying a set of tasks in which stakeholders should be involved, but without relating them to stakeholder attributes. These relate to, among other things, informing: stakeholders are only informed about (planned) activities by politicians and decision makers, but cannot influence the planning process; consultation: decision makers seek discussion with citizens, the results however do not consist of any commitment from the official side; advising: citizens may develop solutions and report problems to decision makers. Their input will be considered by the decision makers, however, they still have the final decision; co-producing: decision makers and citizens jointly agree on issues to be solved and adequate solutions. The decision makers commit themselves to these solutions; co-deciding: decision-making bodies leave the policy planning to the citizens and only keep an advisory role. The results, however, need to be in

line with certain preconditions (policy framework). In our article, we have made the tasks of stakeholders more specific, and moreover, we have linked them to their attributes.

In the literature items indicated, residents are identified as key stakeholders. In our research, residents are also key (albeit unaware and unpredictable) beneficiaries of solutions and should be involved in a number of tasks, especially in giving their opinion on solutions, voicing their needs and evaluating solutions for sustainable flows of people in the city. These stakeholders, however, are not at the forefront of the city's sustainable passenger transport ecosystem, as they do not have significant power of influence, which is reflected, among other things, in the fact that we do not recommend their direct involvement in the creation of the city's strategy. In addition, residents, as an uninformed, unpredictable beneficiary, have a low awareness of the impact of policies that are not geared towards sustainability. Residents focus on the current convenience of moving around in the city rather than the long-term effects. It is therefore more difficult for them to accept solutions that reduce individual car transport in the city and these measures may be inhibited by them. They therefore also become unpredictable in their approach to accepting innovations for sustainable urban flows. Stakeholders such as health-related organisations, companies designing smart and logistical solutions in the city, educational organisations and estate managers were singled out as beneficiaries. This is a very diverse group of stakeholders who experience the benefits of a sustainable flow of people in the city in different ways. By comparison, educational organisations benefit due to the increased safety of children and young people within schools, while businesses involved in designing smart and logistical solutions in the city benefit economically.

The results we obtained are an important addition to previous research on the stakeholders of people flows in the city. In particular, the identification of the leaders of this ecosystem is interesting. Among the leaders in Polish cities, experts identified organisations related to public safety, transport infrastructure managers, micromobility organisations and public transport organisations. These are stakeholder groups that are not recommended in other literature for cooperation in the area of shaping sustainable urban passenger transport systems. In particular, micromobility organisations and public safety organisations are rarely mentioned in the literature. It is worth emphasising again at this point that the research was concerned with the passenger transport ecosystem and the proposed solutions in this area are intended to be oriented towards sustainability. This perspective changes, for many stakeholders, both the direction and strength of influence, as well as the predictability and level of interest. Thus, considering, for example, taxi corporations, if the analysis were to focus on the development of the city's logistics system increasing road capacity (without including the pillars of sustainability), it can be hypothesised that the level of interest of this stakeholder in particular would change to a high level, and furthermore, the strength of influence would not be inhibiting. Similarly, the results obtained for hypermarkets and discount shops and for retail chains can be interpreted. In our research, these organisations were assigned the role of a declared opponent. This role is due to the inhibitory effect on solutions limiting the individual mobility of residents, especially with regard to the use of motor vehicles. It can be assumed that the development of the logistics system itself, increasing road capacity, would be assessed differently.

Recommendations derived from assigning stakeholder characteristics to tasks relate to shaping the city's logistical system in line with the pillars of sustainability. The identification of a group of leaders of such a specified ecosystem is crucial, as leaders should be involved in all tasks. Thus, cooperation with this group should be very precisely designed by the local government units. In this paper, we have not analysed the types of stakeholder relationships according to the assigned role. This is an idea for further research on this topic. Great importance in the results obtained is also attributed to the unpredictable main player. This stakeholder, which in Polish cities along with other cities, is included in as many as seven of the eight tasks. The great importance of cooperation with other cities is also emphasised by other researchers arguing that investments in the flows of both people and cargo very often go beyond the borders of a single city or municipality [82]. The difficulty in demarcating

the boundaries of the impact of investments can be seen especially in metropolises. Thus, the inclusion of other cities in the creation of solutions, as well as their implementation, is in many cases even necessary, and their participation in the creation of the city strategy will allow for a coherent policy of sustainable development of logistics systems in the region. Slightly less importance is attributed to those organisations characterised by the role of patron. Due to the high power of influence of these organisations, they should be included in the creation of the strategy. They do not have a vested interest in the sustainability of the movement of people in the city but support activities in this regard. Thus, they become impartial initiators of measures aimed at sustainable flows. They are valuable partners for the city and should also participate in the promotion of activities and solutions aimed at sustainable development, as well as being an advisory body. In Polish cities, the role of patron should be played by R&D and environmental organisations.

The main limitation of the research results obtained is that the study was conducted only on Polish cities. Thus, the inclusion of the stakeholder in a specific role and task cannot be considered universal. For example, in Polish cities, residents were assessed as an unaware, unpredictable beneficiary, which was due to their low awareness of the consequences of the development of the city's logistical system. One can assume that conducting the same research in, for example, the Netherlands, the results would be quite different. We obtained a universal method for assigning stakeholders to roles and indicating which tasks they should be involved in; however, it is worth conducting a comparative analysis of stakeholder roles in different countries. The second limitation, which is a consequence of the limitation of the first one, is the narrowing down of the group of experts to experts coming from Poland. This was necessary due to the requirement for their detailed knowledge of the solutions found in Polish cities; however, it excluded the selection of experts from the group of global specialists. Thus, we obtained precise knowledge allowing us to assign the stakeholders of Polish cities to roles and tasks, but on the other hand, we limited the analysis at the level of role characteristics and task characteristics. Expanding the group of experts at the level of model characterisation (relating role to predictability, level of interest, direction and strength of influence) will be an important stage of further research. We believe that it is also important to conduct research from the perspective of the stakeholders themselves and, based on the clash between their opinions and the knowledge obtained in this paper, to develop a concept for building relationships with each stakeholder group. The analysis of stakeholder roles in the ecosystem of sustainable urban flows, as we indicated in the introduction, is part of an extensive research on sustainable urban logistics stakeholders. Combining the research results obtained in this paper with those on the sustainable urban freight flow ecosystem and the sustainable urban waste management ecosystem will allow a comprehensive and holistic interpretation of the roles and tasks of each stakeholder.

## 6. Conclusions

The movement of people in a city is a fundamental sub-system of the urban logistics ecosystem. It is strongly influenced by urbanisation processes, changes in people's behaviour and needs and the assumptions of sustainable development. Thus, cities face a major challenge in organising passenger transport that meets people's needs and expectations and ensures a high level of satisfaction, while taking into account both economic and social or environmental aspects. In order to meet this challenge, city managers, who are responsible for developing strategies and policies (including in the area of passenger transport), need to identify and involve different stakeholder groups. This issue formed the basis of our research. As our systematic literature review has shown, the area of urban passenger transport is increasingly being addressed by researchers. This is evident in the increase in publications. At the same time, we have identified a theoretical and cognitive gap related to the issue of urban passenger transport stakeholders. This shows the considerable research potential of the issue we have addressed.

The research we conducted was designed to answer the three research questions posed in the Introduction. To this end, we adopted our proposed research methodology on stakeholder roles in sustainable urban passenger transport. They form the basis for indicating recommendations for city governments to involve different stakeholder groups in tasks aimed at sustainable urban passenger transport. Within the framework of the first research question, we identified 24 stakeholder groups for sustainable urban passenger transport through our expert research on the identification of stakeholder characteristics and their analysis. Within the framework of the second research question, we assigned to all identified stakeholders the roles they should play in terms of sustainable people transport in the city. Of the total 16 roles proposed, we assigned a total of 10 roles to the identified stakeholders. As part of the third adopted question, we studied the involvement of stakeholders in the ongoing tasks related to the development of a sustainable passenger transport policy in the city. All stakeholders were assigned to the eight proposed tasks, taking into account their previously assigned role. Their assignment to each task indicates recommendations for city governments on how to involve stakeholders in shaping a sustainable urban passenger transport policy.

Within the framework of our research, we have identified three main limitations. The methodology we adopted is universal, while the research of stakeholders and the assignment of roles and tasks to them was carried out with reference to Polish cities only. The second limitation is the selection of experts who came only from Poland. This was the result of conducting stakeholder research for Polish cities. The third limitation is the adopted perspective of the stakeholder research. It focuses on the experts' opinions, on the basis of which the stakeholders are assigned roles and tasks they should perform in the area of sustainable passenger transport. A consequence of this limitation, but also a prerequisite for the applicability of the method we have developed, is the appointment of a team of experts from outside the various stakeholder groups. However, the knowledge and experience of the experts should absolutely be based on the knowledge of the environment of a particular stakeholder group in terms of its motivations, resources and styles of operation. In our experience, it is a difficult task to select the right experts, and the effectiveness of the entire research process depends on their proper selection.

The formulated limitations allow us to identify directions for our further research. First of all, it should be emphasised that the problem addressed in the paper is part of a broader analysis of the urban logistics ecosystem and the stakeholders in this ecosystem. As part of these, in addition to the results presented in the paper concerning stakeholders in sustainable flows of people, we are conducting analyses devoted to sustainable flows of goods and sustainable waste management in the city. The results of all three areas will allow us to identify the roles and tasks of the different stakeholders in the entire urban logistics ecosystem. We plan to transfer the research conducted in this area to cities in other countries. This will allow us to conduct a comparative analysis of the roles and tasks of urban logistics ecosystem stakeholders in different countries. In further research, we assume we will broaden and diversify the group of experts by including experts representing other countries in our research. The last direction of our further research is to conduct it from a stakeholder perspective. This will provide us with information on stakeholders' perceptions of their roles and tasks in the urban logistics ecosystem. This perspective will also allow us to compare the results regarding the research on stakeholder roles and tasks in the urban logistics ecosystem.

**Author Contributions:** Conceptualisation, E.P., M.K. and K.D.; methodology, E.P., M.K. and K.D.; software, K.D.; validation, E.P., M.K., K.D.; formal analysis, E.P.; investigation, E.P., M.K. and K.D.; writing—original draft preparation, E.P., M.K. and K.D.; writing—review and editing, E.P., M.K. and K.D.; visualization, E.P., M.K. and K.D. All authors have read and agreed to the published version of the manuscript.

**Funding:** The research presented in the paper was supported by statutory work 13/050/BK/22/0001 carried out at the Faculty of Organisation and Management, Silesian University of Technology.

**Institutional Review Board Statement:** Not applicable.

**Informed Consent Statement:** Not applicable.

**Data Availability Statement:** All data obtained as part of this study have been archived with the authors of the article.

**Conflicts of Interest:** The authors declare no conflict of interest.

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
