# Peer review of "The Role of Stakeholders in Creating Mobility in Logistics Systems of Polish Cities"

_sustainability, doi:10.3390/su15031790_

Round 1

Reviewer 1 Report

I recommend to complete the literature review with more current relevant sources with the topic of the contribution (COVID-19) from the years 2021 - 2022.

In my opinion, part of the discussion  should be part of the results. I recommend adjusting it. The discussion should also focus mainly on a critical evaluation of one's own research results.

In conclusion, there is a lack of critical assessment of the results achieved. 

Author Response

Dear Sir/Madam,

Thank you for taking the time to review our manuscript and all comments! They certainly allowed us to improve our paper. All comments have been thoroughly analysed and entered into the manuscript. 

Authors

Reviewer 2 Report

The authors present a very well structured work dealing with city logistics issues, as well as the role of stakeholders for a sustainable urban mobility. For this aim, three research questions are formulated in the introduction section.

However, some major and minor changes should be undertaken for the authors for the work acceptation. Some of them are the following:

Despite the abstract is really long, it doesn´t present any finding of the work, which could spur the interest for its reading. I recommend to include at least one important finding and to delete the explanation describing the methodology and just naming that the work contains a methodology.

In the abstract is written that one of the aims of the work is to develop recommendations for city governments in terms of stakeholder´s tasks. In my opinion, these recommendations or the most important ones are not clearly reflected in the conclusions of the paper.

There are many keywords and some of them contains several words like “movement of people in the city”. Try both to reduce the number of keywords to a most representative group which represents the research and to shorten if possible some of them.

Although in most of the research papers, the authors do not spend a healthy effort to explain some aspects concerning their work or do not reflect properly what has been done previously in the research field, the authors behave conversely in the presented research. However, in some sections of the paper, there exists an abundance of information which makes the work hard for its reading and distracting the readers. Take a look at section 2.1 with almost 5 pages and also some parts without a full stop (see from line 250 up to 327).  Similar suggestion may be applied to section 5 and 6.

It´s really necessary such long paragraphs? For the sake of clarity and to make more attractive the paper, try to reduce former sections, highlighting key references and deleting useless information.

In line 452 is written “Factors that influence how to work with stakeholders are the level of interest, the direction and strength of the stakeholder's influence on the area and the level of predictability of their behavior”. Is it based on the authors´ opinion? Can you justify it?

One of the most shortcoming of the work agreed by the authors, is that referred to its application to only Polish cities. As a consequence, it´s not a universal study to be extended to any city all over the world. Nevertheless, the title do not reflect this limitation and should be changed, stating this issue.

In line 413 are identified 5 five groups of stakeholders: shippers, freight carriers, administrators, residents and others. These groups really represent the stakeholders regarding urban logistics roles. Nonetheless, experts involved in the work were the only group of stakeholders represented in the study according to the authors. How the results may be affected by this weakness of the research?

Author Response

(The authors gave the same response as above.)

Round 2

Reviewer 2 Report

All my recommendations have been properly addressed. Thanks a lot!